# Structural basis of proton translocation and force generation in mitochondrial ATP synthase

**Niklas Klusch[†], Bonnie J Murphy[†], Deryck J Mills, Özkan Yildiz, Werner Kühlbrandt***

Department of Structural Biology, Max Planck Institute of Biophysics, Frankfurt, Germany

**Abstract** ATP synthases produce ATP by rotary catalysis, powered by the electrochemical proton gradient across the membrane. Understanding this fundamental process requires an atomic model of the proton pathway. We determined the structure of an intact mitochondrial ATP synthase dimer by electron cryo-microscopy at near-atomic resolution. Charged and polar residues of the *a*-subunit stator define two aqueous channels, each spanning one half of the membrane. Passing through a conserved membrane-intrinsic helix hairpin, the lumenal channel protonates an acidic glutamate in the *c*-ring rotor. Upon ring rotation, the protonated glutamate encounters the matrix channel and deprotonates. An arginine between the two channels prevents proton leakage. The steep potential gradient over the sub-nm inter-channel distance exerts a force on the deprotonated glutamate, resulting in net directional rotation.

DOI: https://doi.org/10.7554/eLife.33274.001

*For correspondence:
werner.kuehlbrandt@biophys.
mpg.de

[†]These authors contributed equally to this work

## Introduction

Mitochondrial ATP synthase uses the energy of the electrochemical proton gradient across the inner mitochondrial membrane to produce ATP from ADP and phosphate by rotary catalysis (*Abrahams et al., 1994*; *Gresser et al., 1982*). ATP synthases consist of the catalytic $F_1$ head and the $F_o$ subcomplex in the membrane (*von Ballmoos et al., 2009*). Rotation is driven by protons flowing down the membrane gradient through the $F_o$ subcomplex. Understanding how this fundamental process generates rotary force requires an atomic model of the proton pathway. Until now, no high-resolution structure of an intact, functionally competent mitochondrial ATP synthase has been reported. The recent cryo-EM structure of the $F_o$ subcomplex dimer isolated from yeast mitochondria (*Guo et al., 2017*) indicated the positions of key residues in the proton pathway. We have determined the structure of the complete mitochondrial ATP synthase dimer from the unicellular green alga *Polytomella sp.* Our structure reveals two prominent aqueous channels, each spanning one half of the membrane, that conduct protons to and from the conserved glutamates in the rotor ring. Protonation and deprotonation of these glutamates drives ring rotation and ATP synthesis.

Rotor rings of F-type ATP synthases consist of 8 (*Watt et al., 2010*; *Zhou et al., 2015*) to 15 (*Pogoryelov et al., 2009*) identical *c*-subunits that each form a hydrophobic helix hairpin. Mammalian mitochondria have a $c_8$-ring, while yeasts (*Hahn et al., 2016*; *Stock et al., 1999*) and *Polytomella* (*Allegretti et al., 2015*) have 10 *c*-ring subunits, which we refer to as *c*A to *c*J. A conserved glutamate (*c*Glu111 in *Polytomella*) serves as the *c*-subunit proton-binding site (*Meier et al., 2005*; *Pogoryelov et al., 2009*). The previous 6.2 Å cryo-EM map of the *Polytomella* ATP synthase dimer indicated two long, membrane-intrinsic helix hairpins in subunit *a* (*Allegretti et al., 2015*), but did not resolve sidechains. The helix hairpins run roughly at right angles to the *c*-ring helices. The longest helix bends around the *c*-ring, positioning the strictly conserved *a*Arg239 and other key subunit

*a* residues next to the *c*-subunit protonation site. Subsequent structures of F-type (*Guo et al., 2017*; *Hahn et al., 2016*; *Morales-Rios et al., 2015*) and V-type ATPases (*Mazhab-Jafari et al., 2016*) at 3.7 to 7 Å resolution have shown that the long membrane-intrinsic helix hairpins are a conserved and apparently essential feature of all rotary ATPases (*Kühlbrandt and Davies, 2016*), but the reason for this was not understood until now. Two proton channels were proposed to provide access to the *c*-ring protonation sites (*Vik and Antonio, 1994*) and first observed in the 6.2 Å *Polytomella* structure (*Allegretti et al., 2015*). Like the *a*-subunit helix hairpins themselves, the channels appear to be conserved in all rotary ATPases (*Kühlbrandt and Davies, 2016*).

## Results and discussion

### Structure determination and atomic model

We performed single-particle electron cryo-microscopy (cryo-EM) on ATP synthase dimers of the colourless unicellular alga *Polytomella sp.* to reveal the proton translocation pathway in atomic detail. With a molecular mass of ~1.6 MDa and its bulky peripheral stalk of subunits ASA 1-9 (ATP synthase associated proteins 1–9) (*Figure 1*; *Figure 1—figure supplement 1*) (*Vázquez-Acevedo et al., 2006*), the robust, V-shaped *Polytomella* dimer is well suited to high-resolution cryo-EM.

A total of 90,142 particle images (*Figure 1—figure supplement 2A*) from 9,518 movies recorded with a direct electron detector in counting mode were aligned and refined to yield a map at 4.1 Å resolution (*Table 1*; *Figure 1—figure supplement 2B*), in which most sidechains of the ~14,000 residue dimer are visible (*Figure 1*; *Video 1*), except in the $F_1$ heads and central stalks, where blending of multiple rotational states reduces map definition. Subvolume masking of the bulky peripheral stalk and associated $F_o$ subunits improved the local resolution to 3.7 Å (*Figure 1—figure supplement 2C*). The best-defined region of the assembly includes subunit *a*, the *c*-ring interface and ASA 6, which was built *de novo* (*Figure 2*; *Figure 2—figure supplement 1*).

Sequences of functionally important subunit *a* regions are highly conserved (*Figure 1—figure supplement 3*) and likely to have similar structures. In yeasts and mammals, subunit *a* is mitochondrially-encoded (known as ATP6 in human mitochondria), whereas in *Chlamydomonas reinhardtii* and, presumably, its close relative *Polytomella*, it is nuclear-encoded (*Funes et al., 2002*). N-terminal sequencing indicates that *Polytomella* subunit *a* has a 94-residue mitochondrial targeting sequence (*Vázquez-Acevedo et al., 2006*). The polypeptide forms a total of six α-helices H1 to H6. The mostly hydrophobic residues of the mature gene product are clearly resolved (*Figure 2—figure supplement 1*), except for the 11-residue loop connecting H4 and H5. The N-terminal H1 on the matrix-facing membrane surface is amphipathic, whereas in fungal and mammalian *a*-subunits, the first helix crosses the membrane (*Guo et al., 2017*; *Hahn et al., 2016*; *Zhou et al., 2015*). The four-helix bundle of hairpins H3/H4 and H5/H6 is immersed in the hydrophobic membrane interior. The 52 residues of H5 include the essential (*Mitome et al., 2010*) *a*Arg239 and seven other charged or polar sidechains, interspersed with hydrophobic residues in a striking, strongly conserved pattern (*Figure 1—figure supplement 3*). Our structure puts these residues into a functional context of proton translocation and force generation.

### The lumenal channel conducts protons to the *c*-ring rotor through a helix hairpin in the membrane

Two prominent aqueous channels span half of the $F_o$ assembly, one from each side of the membrane (*Figures 3,4*). The lumenal channel enables proton access to *c*Glu111 from the crista lumen. Its entrance is a 23 by 37 Å funnel between the *c*-ring, the loop connecting the membrane-intrinsic H3/H4 hairpin, and the two trans-membrane helices of subunit ASA 6 (*Figure 3A,C,D*). Although there is no detectable sequence homology, ASA 6 appears to take the place of the peripheral stalk subunit *b* in yeasts and mammals (*Guo et al., 2017*; *Hahn et al., 2016*; *Zhou et al., 2015*). Four lipid acyl chains and densities that accommodate two phosphatidyl head groups are resolved at the rim of the lumenal funnel (*Figure 4—figure supplement 2A*), consistent with a cardiolipin molecule mediating close contacts between ASA 6 and subunit *a* in this position.

In the protein interior, the channel is lined by conserved charged, polar and hydrophobic sidechains (*Figure 4*; *Figure 4—figure supplement 3A*; *Videos 2* and *3*). The lumenal channel extends to a cluster of buried, closely spaced glutamates and histidines (*a*Glu172, *a*His248, *a*His252,

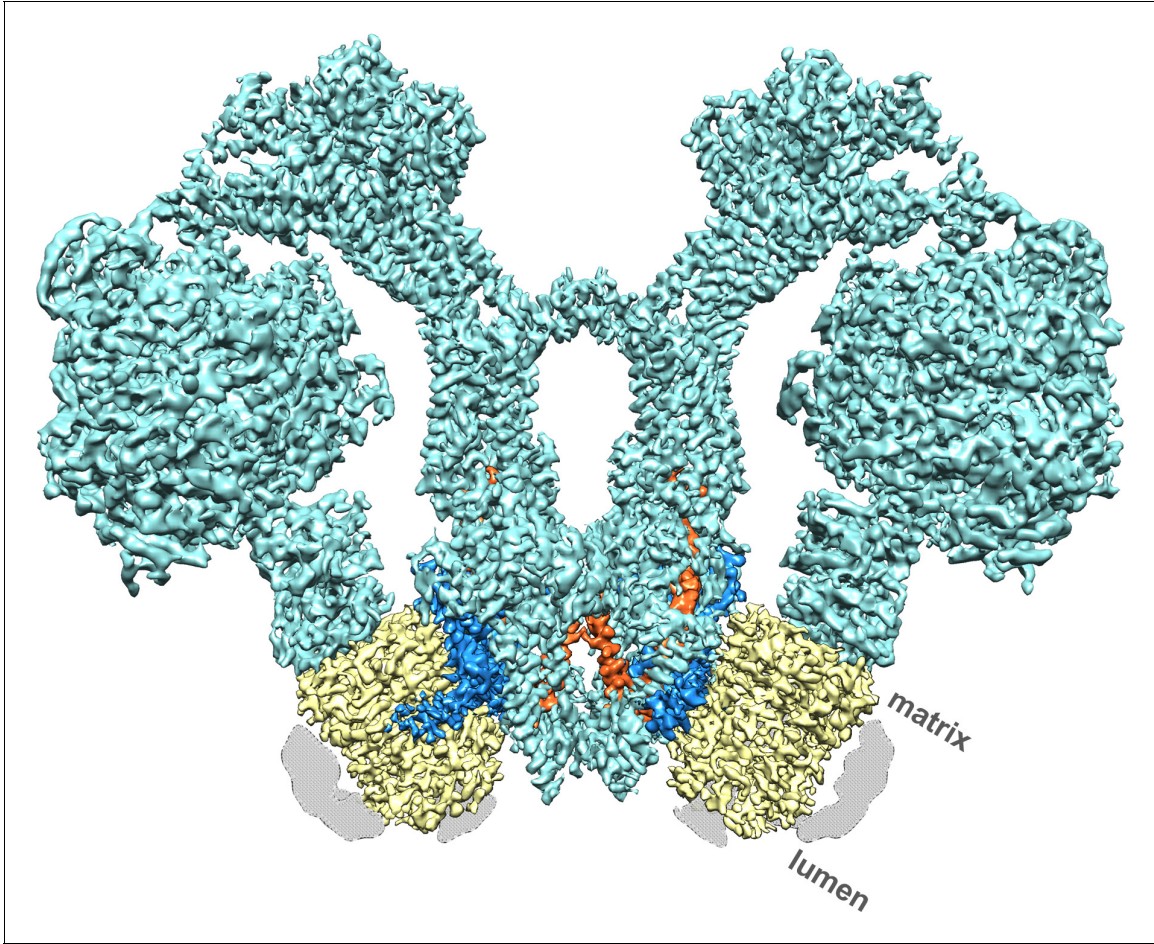

**Figure 1.** Cryo-EM structure of the *Polytomella sp.* F$_1$ F$_o$ ATP synthase dimer at 4.1 Å resolution. Subunit *a*, blue; *c*-ring, yellow; ASA 6, brick; other subunits, cyan; detergent micelle, grey.

DOI: https://doi.org/10.7554/eLife.33274.002

The following figure supplements are available for figure 1:

**Figure supplement 1.** Schematic diagram of ATP synthase dimer from mitochondria of *Polytomella sp.*
DOI: https://doi.org/10.7554/eLife.33274.003
**Figure supplement 2.** Cryo-EM of *Polytomella sp.*ATP synthase dimer.
DOI: https://doi.org/10.7554/eLife.33274.004
**Figure supplement 3.** Subunit *a* sequence alignment.
DOI: https://doi.org/10.7554/eLife.33274.005

*a*Glu288; *Figure 4A*; *Figure 5A*; *Videos 2* and *3*) that appears to serve as a local reservoir for protons to be fed to the *c*-ring glutamates. At *a*Glu288 about 20 Å below the lumenal membrane surface, the channel narrows to 4 by 5 Å and changes direction by 90° towards the *c*-ring (*Figure 3D*; *Figure 4A*). In the hydrophobic membrane interior, the strictly conserved polar sidechains of *a*Asn243 (H5) and *a*Gln295 (H6) that face one another would stabilise the H5/H6 hairpin (*Figure 5B*). The channel passes through the hairpin at the small, conserved sidechains of *a*Ala246, *a*Gly247 (H5) and *a*Ala292 (H6) (*Figure 3C,D*; *Figure 4A*; *Figure 5B*; *Figure 1—figure supplement 3*). Bulky hydrophobic sidechains close by on H5 and H6 keep the helices apart and the channel open (*Figure 4A*; *Figure 5B*). From *a*Glu288 the proton may jump to either of two *c*-ring glutamates. The path to *cA*Glu111 is 4 Å longer than the ~12 Å path to *cB*Glu111 (*Figure 5C*) but includes the hydrophilic sidechains of *a*Asn243 and *cB*Ser112. Therefore, in our static structure the path to *cA*Glu111 is more favourable for proton transfer. In a rotating *c*-ring, the estimated minimum distance from *a*Glu288 to *c*Glu111 is ~11 Å. Note that these distance estimates are subject to an error margin of 3

**Table 1.** Cryo-EM data collection parameters, image processing, and refinement statistics.

**Data collection**

| | |
|---|---|
| Electron Microscope | JEOL JEM-3200FSC |
| Camera | K2 Summit |
| Voltage | 300 kV |
| Energy filter slit width | 20 eV |
| Nominal Magnification | 30,000 x |
| Calibrated physical pixel size | 1.12 Å |
| Pixel size after mag. distortion corr. | 1.105 Å |
| Total exposure | 82.5 e⁻/Å² |
| Exposure rate | 11.5 e⁻/(pixel x s) |
| Number of frames | 45 |
| Defocus range | −0.4 to −5 µm (95% between -0.9 and -2.5 µm) |

**Image Processing**

| | |
|---|---|
| Motion correction software | Unblur/MotionCor2 (with mag. distortion corr.) |
| CTF estimation software | CTFFind4, Gctf (for per-particle CTF) |
| Particle selection software | e2boxer (EMAN2) |
| Micrographs used | 9,518 |
| Particles selected | 117,281 |
| 3D map classification and refinement software | Relion2 |
| Particles contributing to final map | 90,142 |
| Applied symmetry | C2 |
| Global resolution (FSC = 0.143) | 3.68 Å |
| Applied B-factor | -125 Å |

**Model Building**

| | |
|---|---|
| Modeling software | Coot |
| Refinement software | Phenix (phenix.real_space_refine) |
| Number of residues built | 1,039 |
| RMS (bonds) | 0.01 Å |
| RMS (angles) | 1.09° |
| Ramachandran outliers | 0.0% |
| Ramachandran favoured | 94.88% |
| Rotamer outliers | 0.27% |
| Clashscore | 7.5 |
| EMRinger score | 1.42 |

DOI: https://doi.org/10.7554/eLife.33274.008

to 4 Å, since the carboxyl groups of the glutamate sidechains are not visible in the map, as is usual in high-resolution cryo-EM structures due to radiation damage (*Allegretti et al., 2014*).

The strictly conserved *a*Arg239 is positioned halfway between the matrix and lumenal channels (*Figures 3A–C,4*; *Video 2*), forming a positively charged seal that prevents proton leakage from the lumen to the matrix. The map density for this arginine sidechain is particularly well defined (*Figure 2—figure supplement 1*). Its position and orientation do not suggest a salt bridge with the deprotonated *c*-ring glutamate, which would impede ring rotation. *a*Asn243, *c*Ser112, the protonated *c*Glu111 and the two aqueous channels provide a local hydrophilic environment. The shortest distance between the lumenal and matrix channels on either side of *a*Arg239 is ~6 Å.

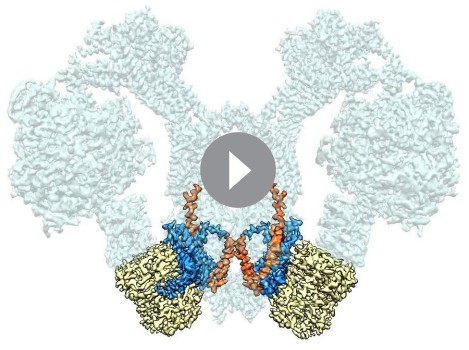

**Video 1.** Three-dimensional map of the 1.6 MDa mitochondrial ATP synthase dimer from *Polytomella sp.* showing the two *c*-rings (yellow), subunits *a* (blue) and ASA 6 (brick). The remaining eight ASA subunits in the peripheral stalks, the central stalks and catalytic F₁ heads are shown in transparent cyan.
DOI: https://doi.org/10.7554/eLife.33274.009

## The matrix channel at the subunit *a/c* interface forms the proton exit pathway

The matrix channel (*Figure 3B,C,E, 4C*; *Figure 4—figure supplement 3B*; *Videos 2* and *4*) is defined by the N-terminal half of H5 (residues 221–239), the C-terminal half of H6 (301–312) and residues *c*Tyr102 to *c*Glu111 of the outer helices of *c*-subunits *A* and *J*. Its deepest point is a 4 Å by 7 Å cavity next to *cJ*Glu111, ~25 Å below the *c*-ring surface. The channel widens to 7 by 13 Å at *a*Arg232 (H5) and *cJ*Leu104 and continues straight to the 36 by 30 Å exit funnel on the matrix side. H4 and H5 describe two sides of a triangle that is wedged open by the conserved aromatic sidechains of *a*Trp189 (H4) and *a*Tyr229 (H5), which forms a hydrogen bond to *a*Thr193 (H4) (*Figure 5D*). The aromatic wedge of *a*Trp189 and *a*Tyr229 induces a change in direction of H5 to follow the curvature of the *c*-ring. Like the lumenal channel, the matrix channel includes conserved charged and polar sidechains (*Figure 4C*; *Figure 1—figure supplement 3*; *Figure 4—figure supplement 3B*), notably *a*Glu225 (H5) and *a*Glu309 (H6), which form a salt bridge with *a*Arg232 (H5). At a distance of 7.4 Å, *a*Glu225 is in a good position to receive protons from *c*Glu111. The protons pass via *a*Glu309 and *a*His312 near the channel exit into the matrix.

The relevance of residues at the matrix channel to human health is highlighted by a number of mutations in ATP6, the *a*-subunit of human ATP synthase, that result in severe and, at present, incurable diseases. Several of these mutations map to H4, H5 and H6 (*Figure 5—figure supplement 1*). Sequence comparison (*Figure 1—figure supplement 3*) indicates that functionally important subunit *a* residues are conserved in ATP6. A change of *atp6*Leu156 (*a*Leu236 in *Polytomella*) to Arg or Pro reduces ATP production by 70%. Both mutations result in Maternally Inherited Leigh Syndrome (MILS) or in Neuropathy, Ataxia and Retinitis Pigmentosa (NARP) Syndrome (*Cortés-Hernández et al., 2007*; *Holt et al., 1990*; *Kucharczyk et al., 2009*). *a*Leu236 marks the point where H5 bends around the *c*-ring. A change of the nearby Trp (*a*Trp189 in *Polytomella*) in H4 to arginine causes Bilateral Striatal Lesions (*De Meirleir et al., 1995*). *a*Trp189 appears to be crucial for keeping H4, H5 and H6 apart and the matrix channel open. Replacing these residues by an arginine or proline would disrupt the interaction of H5 with the *c*-ring rotor, impairing proton translocation and ATP synthesis. Mutations of *atp6*Leu217, *atp6*Leu220 and *atp6*Leu222 (*a*Leu302, *a*Val305 and *a*Val307 in *Polytomella*) on H6 in the same region of the long helix hairpin also result in Leigh Syndrome and reduced ATPase activity (*Castagna et al., 2007*; *Moslemi et al., 2005*; *Thyagarajan et al., 1995*) (*Figure 5—figure supplement 1*). Our structure thus provides direct new insights into the cause of serious mitochondrial diseases.

## Mechanism and energetics of ring rotation

Rotation of the *c*-ring is driven by the proton-motive force (pmf) across the inner mitochondrial membrane, which has a chemical component ($\Delta$pH $\cong$ 0.8 units, equivalent to ~50 mV) and an electrostatic component ($\Delta\Psi \cong$ 150 mV). The higher concentration of protons acts to protonate *c*-subunit *A* in the lumenal channel, whereas in the matrix channel, the incoming protonated *c*-subunit *J* would lose its proton to the pH 8 matrix (*Figure 3A,B*; *Figure 6*; *Video 2*). Protonation in the lumenal channel neutralises the residue and favours the buried sidechain conformation observed in x-ray structures of isolated *c*-rings (*Pogoryelov et al., 2010*). This would allow *c*-subunit *A* to move to position *B,* where it partitions into the hydrophobic environment of the membrane by counter-clockwise rotation, as seen from the matrix. At the same time, deprotonation of *c*-subunit *J* in the matrix channel

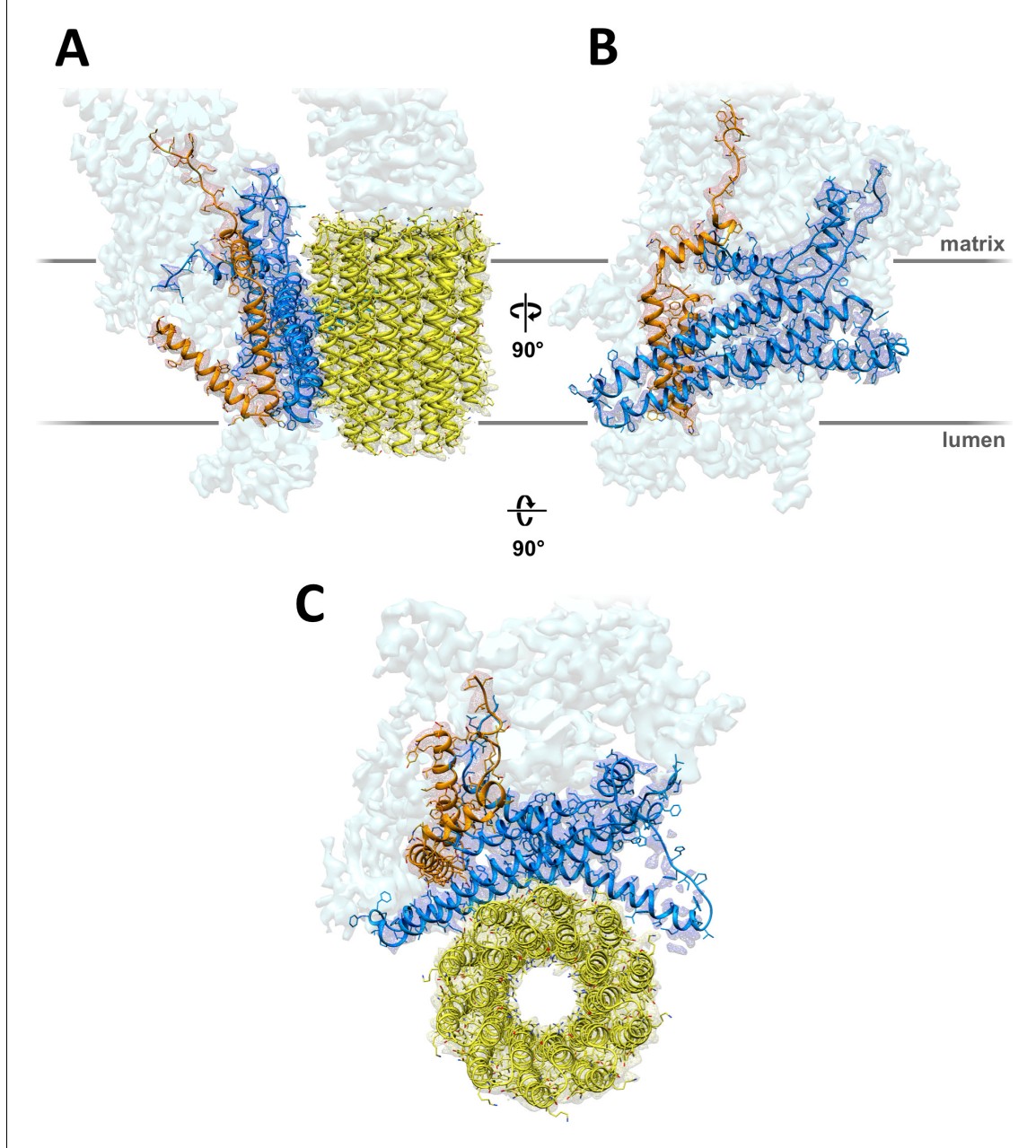

**Figure 2.** 3.7 Å map of *Polytomella* F$_o$ subcomplex with fitted atomic models. (A) Side view with fitted subunit *a*, trans-membrane helices of ASA 6 and *c*-ring. (B) Subunit *a* and ASA 6 seen from the *c*-ring. (C) subunit *a*, *c*-ring and ASA 6 seen from the matrix. Subunit *a*, blue; *c*-ring, yellow; ASA 6, brick; other subunits, light cyan.

DOI: https://doi.org/10.7554/eLife.33274.006

The following figure supplement is available for figure 2:

**Figure supplement 1.** 3.7 Å map with fitted atomic models.

DOI: https://doi.org/10.7554/eLife.33274.007

renders its glutamate negatively charged, attracting it to the positively charged *a*Arg239 halfway between the channels.

In a minimal model of rotary ATPases, torque is generated simply by stochastic bidirectional movement of the rotor, biased by the free energy inherent in the pmf (*Junge et al., 1997*). Other authors propose that the electrostatic field between the channels acts on the negatively charged glutamate, causing the ring to rotate (*Miller et al., 2013*). The atomic coordinates of our structure

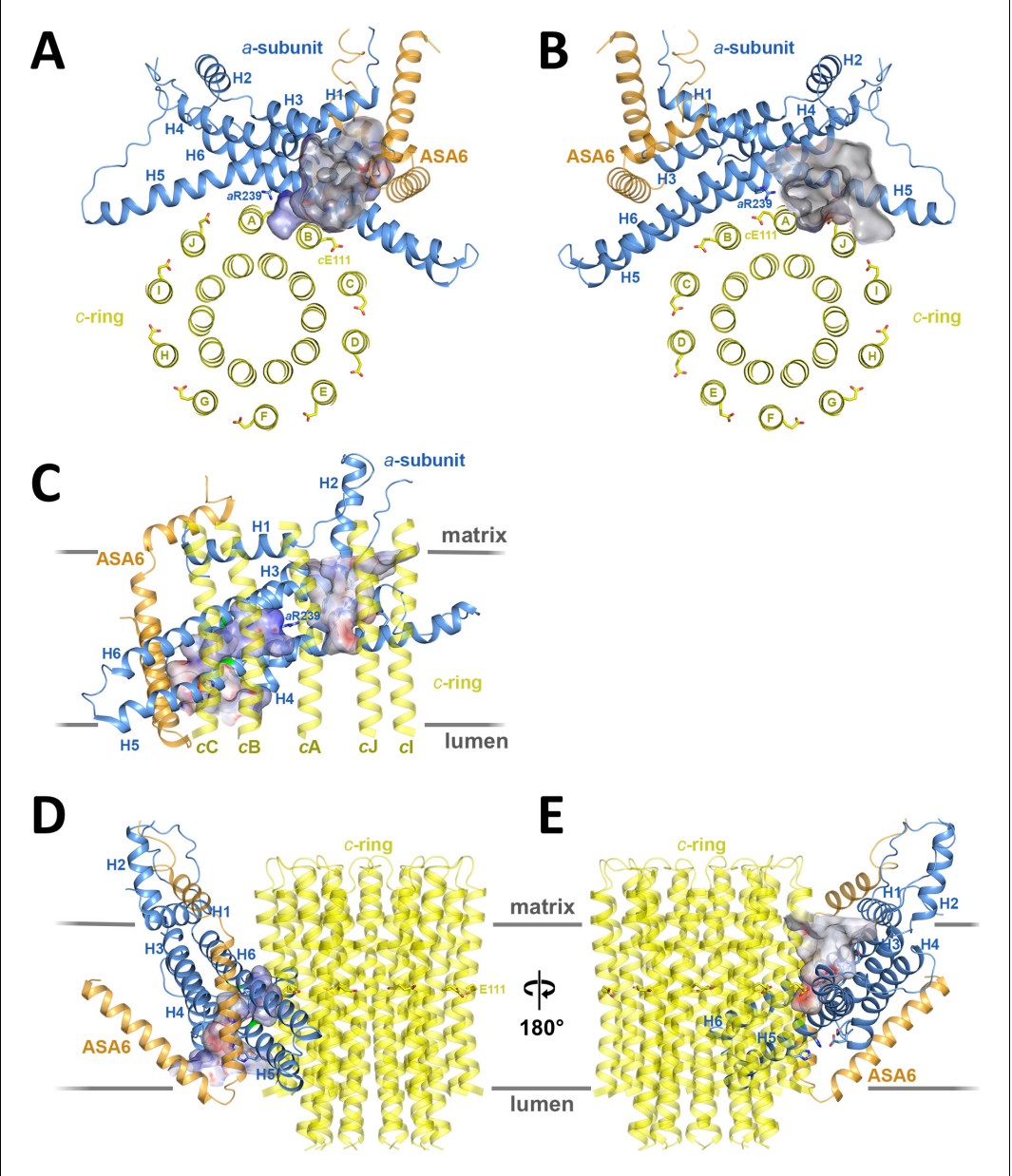

**Figure 3.** Two aqueous channels in $F_o$. (**A**) Lumenal channel seen from the crista lumen. (**B**) Matrix channel seen from the matrix. (**C**) Side view of both channels seen from the *c*-ring, with outer *c*-ring helices in transparent yellow. Lumenal channel, left; matrix channel, right. The strictly conserved *a*Arg239 in H5 separates the lumenal and matrix channels. (**D**) The lumenal channel passes through the H5/H6 hairpin at the small sidechains *a*Ala246, *a*Gly247 (H5) and *a*Ala292 (H6) (green). (**E**) H4, the N-terminal half of H5 and the connecting H4/H5 loop at the matrix channel. Subunit *a*, blue; $c_{10}$-ring, yellow; ASA 6, brick. Channels are shown as potential surfaces (red, negative; blue, positive; grey, neutral). (**A**) and (**B**) display a 5 Å slice of the $c_{10}$-ring at the level of the protonated *c*Glu111.

DOI: https://doi.org/10.7554/eLife.33274.010

allow us to quantify this field. In a vacuum, a potential difference of 200 mV over the minimum 6 Å distance between the channels would generate a local electrostatic field of 330 million V/m, in a direction parallel to the membrane plane. Given a *c*Glu111-*c*Glu111 *c*-ring diameter of 4.2 nm, this electrostatic field would exert a torque of 110 pN nm on the single negative charge of the deprotonated *c*Glu111. The local dielectric of the protein reduces this value by more than 50%, in good

agreement with the experimentally determined torque generated by the $F_1$ head of *E. coli* ATPase in ATP hydrolysis mode of 40 to 60 pN nm (*Kinosita et al., 2000*; *Pänke et al., 2001*;

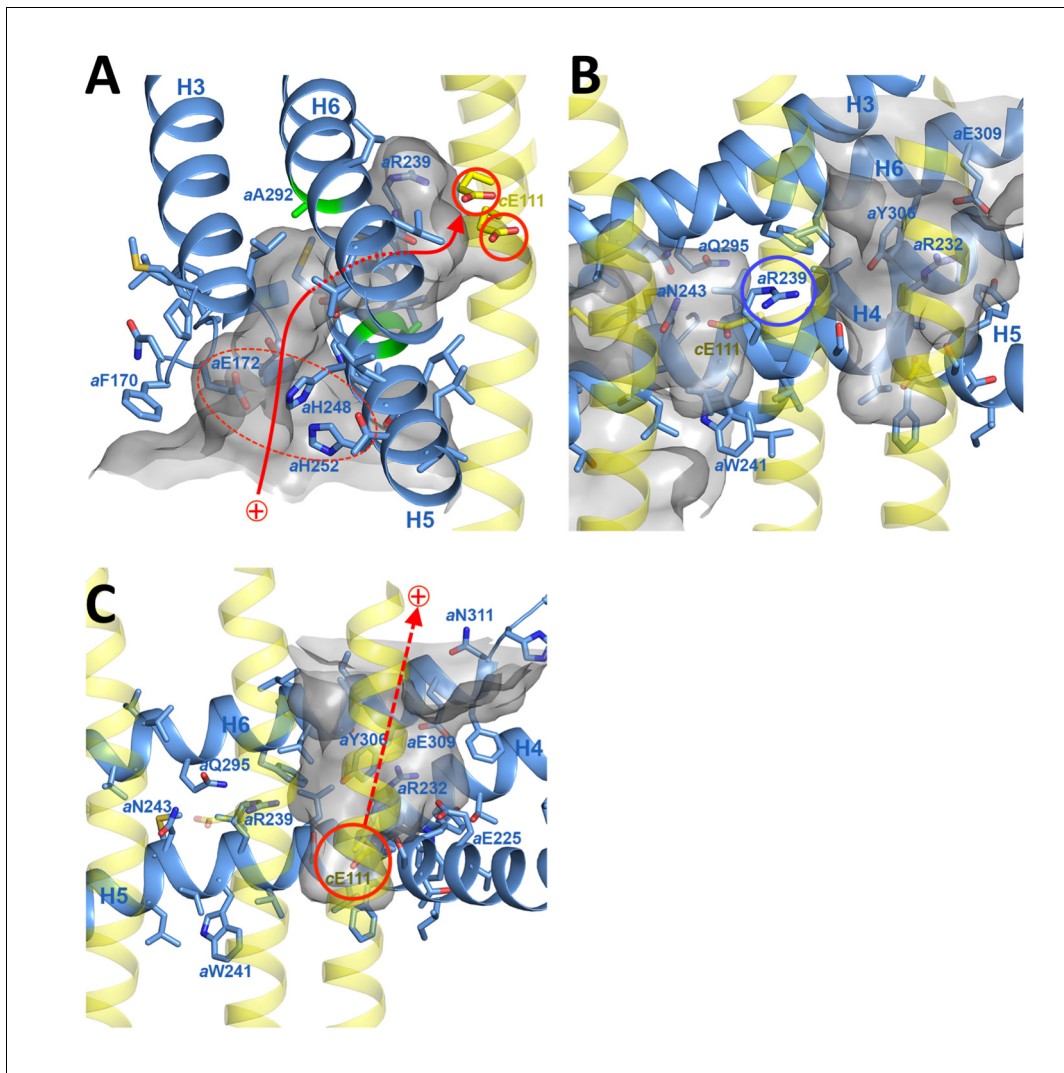

**Figure 4.** Proton pathway through the $F_o$ subcomplex. (**A**) In the lumenal channel, protons (red arrow) pass via the local proton reservoir of *a*Glu172, *a*His248, *a*His252 and *a*Glu288 (dashed red ellipse) through the H5/H6 helix hairpin at the small sidechains of *a*Ala246, *a*Gly247 (H5) and *a*Ala292 (H6) (green) to *c*Glu111 in the rotor ring *c*-subunits (red circles). (**B**) *a*Arg239 (blue circle) is located halfway between the lumenal channel on the left and the matrix channel on the right, forming a seal to prevent proton leakage. *c*-ring helices (transparent yellow) with *c*Glu111 are seen in the foreground. (**C**) In the matrix channel, protons (dashed red arrow) can pass straight from the deprotonated *c*Glu111 to the pH 8 matrix. Subunit *a*, blue; adjacent *c*-ring helices, transparent yellow; aqueous channels, translucent grey; residues in stick representation. *Figure 4—figure supplement 1* shows the fitted model together with the map density in stereo.

DOI: https://doi.org/10.7554/eLife.33274.011

The following figure supplements are available for figure 4:

**Figure supplement 1.** Stereo diagrams of map density (blue mesh) with fitted atomic models.
DOI: https://doi.org/10.7554/eLife.33274.012

**Figure supplement 2.** Stereo diagrams of bound lipid detergents.
DOI: https://doi.org/10.7554/eLife.33274.013

**Figure supplement 3.** Stereo diagrams of channel-lining residues seen from the protein interior.
DOI: https://doi.org/10.7554/eLife.33274.014

*Spetzler et al., 2006*). Considering that, on the molecular scale, catalysis in the ATP synthase must be reversible, the estimated torque generated by the transverse electrostatic field between the two aqueous channels is in the expected range for ATP synthesis, though this does not exclude the possibility that thermal energy plays a role in overcoming activation barriers to rotation. Note that, without the channels, the electrostatic field across the ~35 Å hydrophobic membrane core would be about six times weaker and in the wrong direction (perpendicular to the membrane plane), and hence not able to drive the production of ATP by rotary catalysis. Our structure, in particular the small distance between channels, supports the notion that the force generated by the electrostatic field acting on the deprotonated *c*Glu111 gives rise to directional *c*-ring rotation. The structure of the *Polytomella* F$_o$ ATP synthase thus explains how electrochemical energy is converted into the

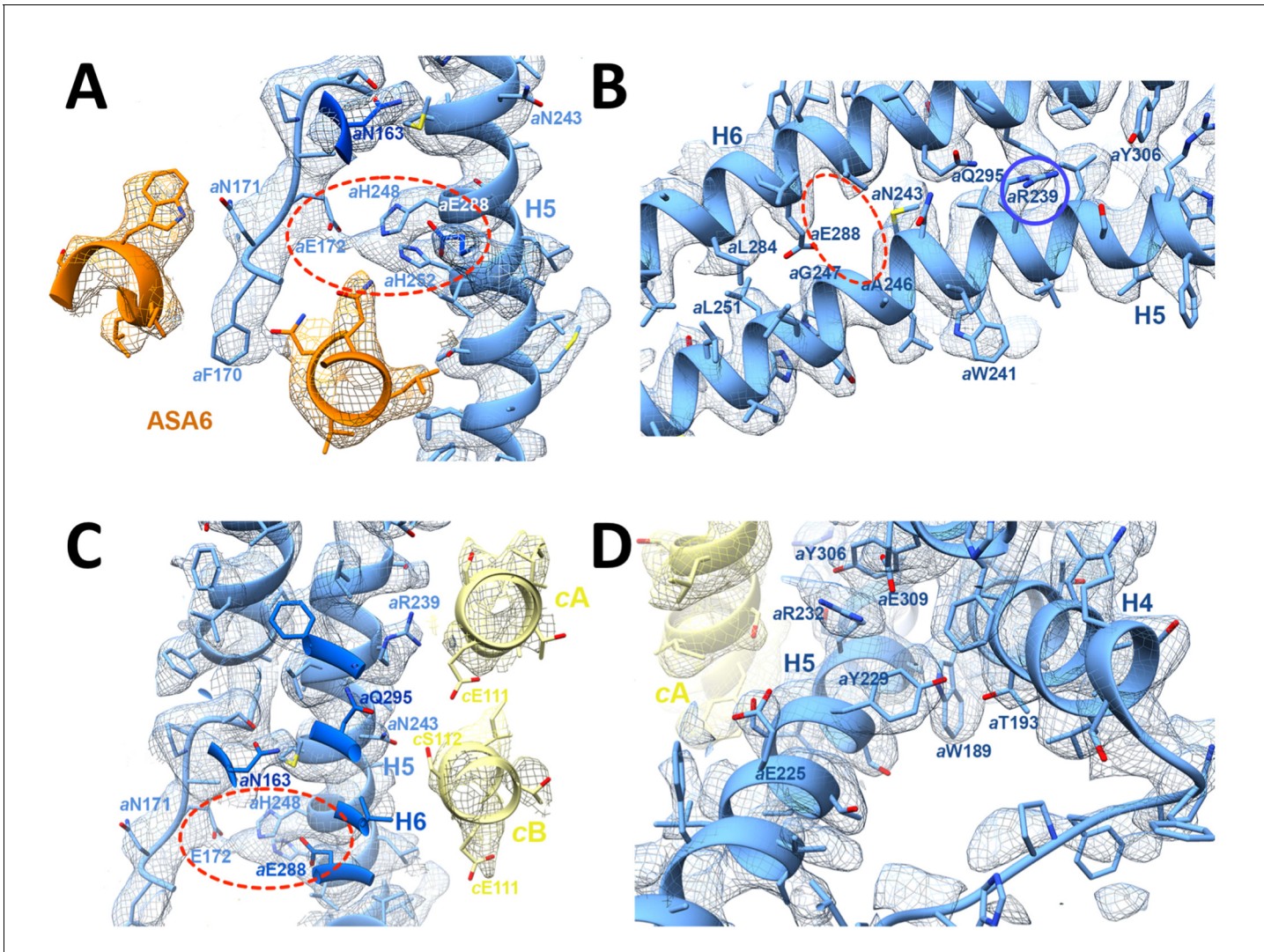

**Figure 5.** 3.7 Å map of functionally important *a*-subunit residues with fitted atomic model. (**A**) Proton reservoir formed by *a*Glu172, *a*His248, *a*His252, *a*Glu288 (dashed red ellipse) in the lumenal channel; (**B**) Interaction of *a*Asn243 (H5) and *a*Gln295 (H6) stabilises the H5/H6 hairpin. The space between *a*Glu288 (H6) and *a*Asn243 (H5) marks the lumenal channel (dashed red ellipse). (**C**) Protons in the lumenal channel can pass from *a*Glu288 to *c*Glu111 of *c*-subunit A near *a*Arg239 (H5) via *c*Ser112 and *a*Asn243. (**D**) *a*Trp189 (H4), *a*Thr193 (H4) and *a*Tyr229 (H5) act as wedges between H4 and the N-terminal end of H5, forming two sides of a triangle.

DOI: https://doi.org/10.7554/eLife.33274.015

The following figure supplement is available for figure 5:

**Figure supplement 1.** Disease-relevant residues in subunit *a*.
DOI: https://doi.org/10.7554/eLife.33274.016

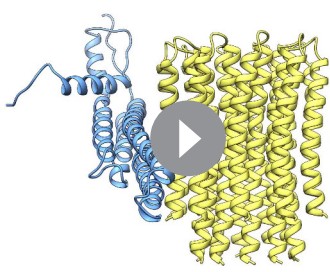

**Video 2.** Three-dimensional arrangement of subunit *a* (blue), *c*-ring (yellow) with lumenal channel in pink and matrix channel in light blue.
DOI: https://doi.org/10.7554/eLife.33274.017

mechanical torque that powers ATP synthesis, as one of the most fundamental life processes. Theoretical studies, made possible by the atomic coordinates that are now available, will be necessary to evaluate the energetics of ATP synthesis in detail.

The hydrophobic membrane environment (circular arrow in *Figure 6*) disfavours movement of negatively charged, deprotonated *c*-ring subunits via the longer of two possible routes between the channels. Conversely, the short route past *a*Arg239 at the *a/c* interface must discriminate against the passage of a protonated *c*-ring subunit. Failure to do so would result in proton leakage and dissipation of the pmf. *a*Arg239 is bound to play a key role in this process. Conceivably, the positive charge on the flexible tether of the arginine sidechain associates with the deprotonated *c*Glu111 on its passage between the aqueous channels, smoothing the energy profile of *c*-ring rotation.

## Conclusion

We determined the structure of a 1.6 MDa mitochondrial $F_1F_o$ ATP synthase dimer by single-particle electron cryo-microscopy. At 3.7 Å resolution, all $F_o$ subunits, helices, loops, most sidechains and some lipids are well resolved. Two prominent aqueous channels defined by subunit *a* and the 10-subunit *c*-ring rotor conduct protons to protonate and deprotonate a *c*-subunit glutamate in the middle of the membrane. Protons enter from the ~pH 7.2 crista lumen through the wide funnel-like opening of the lumenal channel that is lined by conserved polar or charged subunit *a* residues. A

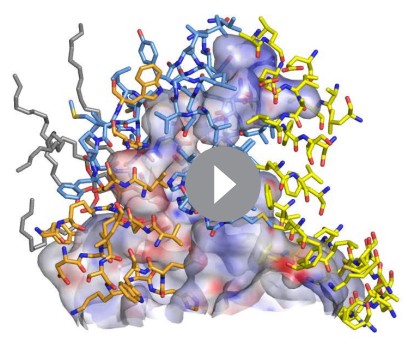

**Video 3.** Arrangement of channel-lining sidechains for the lumenal channel. Sidechains in stick representation are coloured as: subunit *a*, blue; *c*-ring, yellow; ASA 6, brick; lipids, grey. Channels are shown as potential surfaces (red, negative; blue, positive; grey, neutral).
DOI: https://doi.org/10.7554/eLife.33274.018

cluster of glutamates and histidines in the channel about 20 Å below the membrane surface serves as a local proton reservoir. At this point, the channel narrows and turns by 90° towards the *c*-ring, passing between the long, membrane-intrinsic subunit *a* helices H5 and H6. The channel ends at *c*Glu111 of the proximal *c*-ring subunit. *c*Glu111 is protonated and partitions into the hydrophobic membrane environment. Upon a ~320° revolution of the *c*-ring, the protonated *c*Glu111 encounters the aqueous matrix channel defined by a triangle of the membrane intrinsic helices H4, H5 and H6, and the proton escapes to the pH 8 matrix. The positively charged, strictly conserved *a*Arg239 in H5 separates the lumenal and matrix channels, preventing proton leakage. The sub-nm distance between the aqueous channels results in a steep potential gradient, acting on the deprotonated *c*-ring glutamate to generate net directional rotation. The structure explains the fundamental process that drives ATP synthesis in all forms of life. Our atomic model will be essential in evaluating the energetics of proton translocation and force generation in ATP synthases.

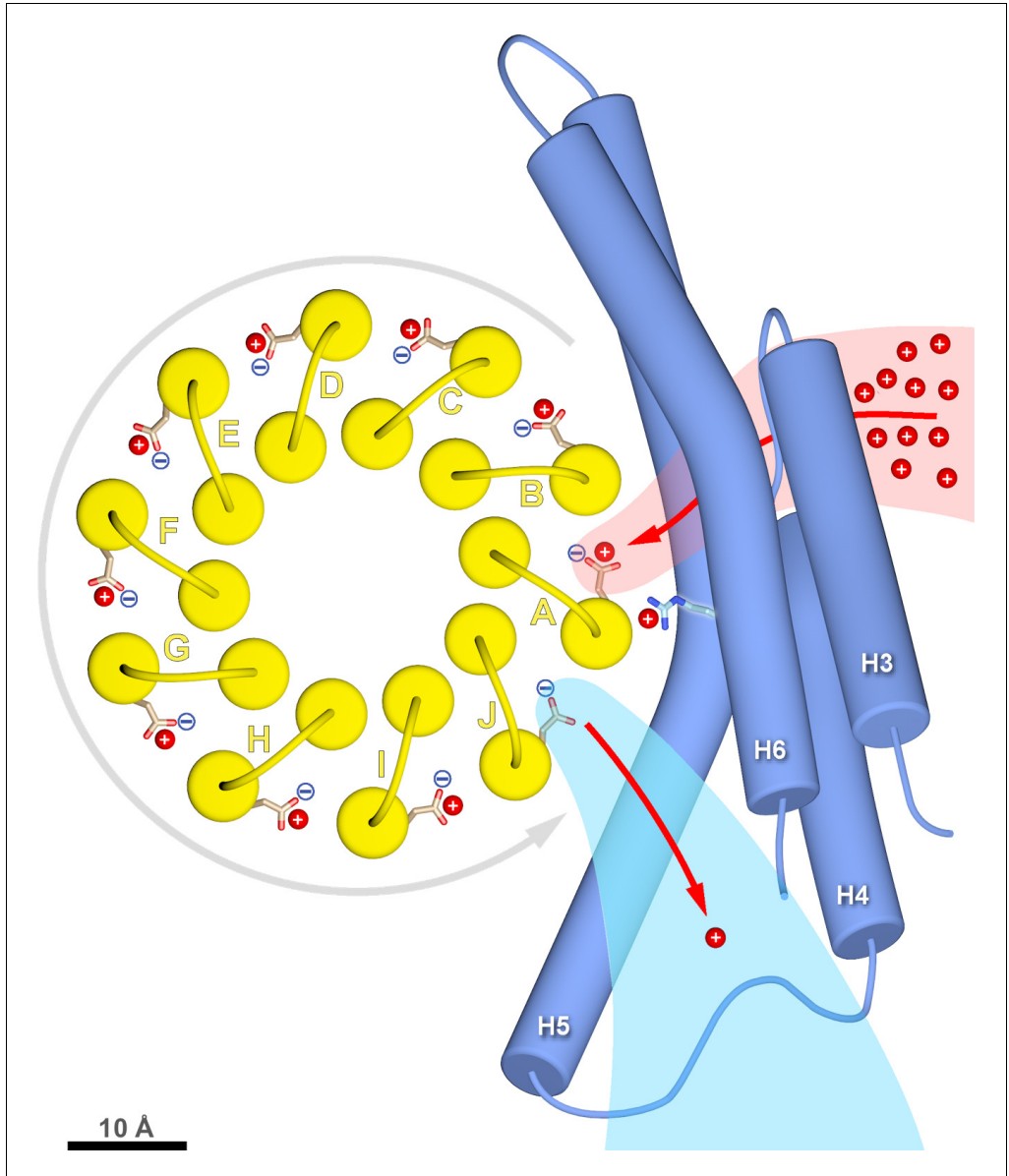

**Figure 6.** *c*-ring rotation is powered by the potential gradient between the lumenal channel (pink) and matrix channel (light blue). The *c*-ring (yellow) and the membrane-intrinsic four-helix bundle of subunit *a* (blue) drawn to scale as seen from the matrix. Protons (red) pass from the crista lumen below the projection plane through the lumenal channel between H5 and H6 to protonate *c*Glu111 of *c*-subunit *J*, while *c*-subunit *J* is deprotonated by the higher pH of the matrix channel. The positively charged *a*Arg239 is likely to interact with the deprotonated *c*Glu111 during its short passage to the lumenal channel. The lumenal and matrix channels approach one another to within 5–7 Å. A pmf of 200 mV between the closely spaced channels creates a local electrostatic field in the range of 40 million to 100 million V/m, depending on the protein dielectric. The field exerts a force on the deprotonated *c*Glu111 that results in net counter-clockwise rotation of the *c*-ring (grey arrow). Scale bar, 10 Å.
DOI: https://doi.org/10.7554/eLife.33274.020

## Materials and methods

Cultures of *Polytomella sp.* (198.80, E.G. Pringsheim) from the alga collection at the Sammlung Algenkulturen Göttingen, Germany were grown aerobically with agitation at room temperature (23 ± 2°C) in MAP medium (*Atteia et al., 2000*). Mitochondria and mitochondrial ATP synthase dimers were isolated as described (*Allegretti et al., 2015*; *van Lis et al., 2005*) with modifications. Briefly,

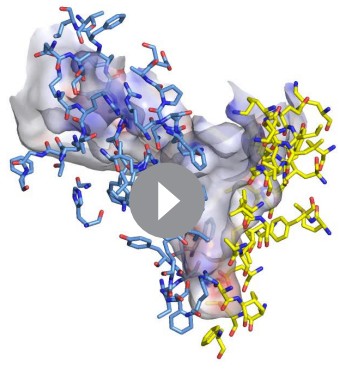

**Video 4.** Arrangement of channel-lining sidechains for the matrix channel. Sidechains in stick representation are coloured as: subunit *a*, blue; *c*-ring, yellow; ASA 6, brick. Channels are shown as potential surfaces (red, negative; blue, positive; grey, neutral).
DOI: https://doi.org/10.7554/eLife.33274.019

mitochondrial membranes (50 mg) were resuspended in solubilisation buffer (10 mM Tris-HCl, pH 8.0, 1 mM MgCl$_2$, 50 mM NaCl, 2% (w/v) n-dodecyl-β-D-maltoside (DDM)) in a total volume of 5 ml. After 30 min at 4°C, unsolubilised material was removed by centrifugation at 20,000 x g for 15 min at 4°C and the supernatant was loaded onto a POROS GoPure HQ column equilibrated in buffer A (10 mM Tris-HCl, pH 8.0, 1 mM MgCl$_2$, 50 mM NaCl and 0.015% (w/v) DDM) on an Äkta purifier (GE Healthcare). The column was washed with buffer A + 100 mM NaCl and ATP synthase dimers were eluted with a linear gradient of 100 mM to 300 mM NaCl in buffer A. Fractions containing ATP synthase dimers were pooled, concentrated to 50 μl in Vivaspin 500 columns with 100,000 molecular weight cutoff and loaded onto a Superose-6 size exclusion column (PC 3.2/30) equilibrated in buffer B (10 mM Tris-HCl, pH 8.0, 1 mM MgCl$_2$, 20 mM NaCl, 0.05% (w/v) DDM) on an Ettan purifier (GE Healthcare). Fractions containing ATP synthase dimers were collected. 250 μM of the substrate analogue AMP-PNP, 5 μM ADP and 0.02% (w/v) sodium azide were added 30 min before freezing to inhibit rotation.

For cryo-EM, 3 μl aliquots of a 1.5 mg/ml inhibited ATP synthase dimer solution were applied to C-flat-MH-4C grids with tobacco mosaic virus (TMV) for spreading, or to Quantifoil R2/2 grids covered with self-perforating hydrogel nanomembranes without TMV (*Scherr et al., 2017*). Grids were plunge-frozen in a Vitrobot (FEI) at 70% humidity, 10°C with 5.5 s (CF-MH-4C) or 9 s (hydrogel membranes) blotting time.

Images were acquired in electron counting mode with a K2 Summit direct electron detector on a JEM-3200FSC field emission cryo-TEM (JEOL, Tokyo) with in-column electron energy filter (slit width 20 eV) at 300kV. The nominal magnification was 30,000x, resulting in a specimen pixel size of 1.12 Å. 45-frame dose-fractionation movies were recorded manually at a defocus range of −0.4 to −5.0 μm (with 95% of micrographs being in the range −0.9 to −2.5 μm) with 0.2 s per frame at an electron flux of 11.5 e⁻/pixel/s.

Specimen movement between movie frames was corrected using Unblur (*Brilot et al., 2012*), followed by MotionCor2 with magnification distortion correction (*Zheng et al., 2017*), resulting in a corrected pixel size of 1.105 Å. Combining these two programs gave better results than either program alone, as judged by quality of a 3D reconstruction for a subset of the micrographs. ATP synthase dimers were picked manually with e2boxer. Micrograph CTF parameters were calculated using CTFFind4.1.5 (*Mindell and Grigorieff, 2003*), and per-particle defocus values were refined using gctf (*Zhang, 2016*). Further processing was carried out in Relion2.0 (*Kimanius et al., 2016*; *Scheres, 2012*). Dimer images were extracted and refined to a 40 Å lowpass-filtered reference with a soft mask surrounding the entire dimer. Particle polishing was carried out, and polished images were classified in 3D. Four of five classes (90,142 of 117,281 particles) were selected and combined for further refinement with a soft mask surrounding the dimer, yielding an overall resolution of 4.14 Å. Focussed refinement using a mask around the peripheral stalk and Fo region gave a better-resolved map of this region (3.68 Å resolution). The map was sharpened using a B-factor of −125 Å$^2$ and low-pass filtered to 3.68 Å with the automated post-processing utility. Local resolution was calculated with the LocalRes utility in Relion.

The quality of the cryo-EM map of the *Polytomella* ATP synthase dimer enabled manual *de novo* modelling of all subunits. For the *c*-ring rotor an initial model was based on the structure of a c$_{10}$-ring in the proton-unlocked state at pH 8.3 (pdb code 3U2F). Each *c*-subunit was fitted as a rigid

body in UCSF Chimera (*Goddard et al., 2007*). The model was fitted and built manually in Coot (*Emsley and Cowtan, 2004*) with additional rounds of real-space refinement in PHENIX (*Adams et al., 2010*) (*Table 1*). The final model was converted into a density map using the molmap command in UCSF Chimera to calculate the map-to-model FSC (*Figure 1—figure supplement 2C*). Water-accessible channels were traced with the program Hollow (*Ho and Gruswitz, 2008*).

### Data availability

The cryo-EM map has been deposited to the Electron Microscopy Data Bank (accession number EMD-4176). Atomic models have been deposited to the Protein Data Bank (accession number 6F36).

## Acknowledgements

We thank Gerhard Hummer for discussion and Janet Vonck and Alexander Hahn for reading the manuscript. This work is supported by the Max Planck Society, DFG SFB 807 and an EMBO long-term fellowship to BJM (ALTF 702–2016).

## Additional information

### Competing interests

Werner Kühlbrandt: Reviewing editor, *eLife*. The other authors declare that no competing interests exist.

### Funding

| Funder | Grant reference number | Author |
| --- | --- | --- |
| Max-Planck-Gesellschaft | | Niklas Klusch<br>Bonnie J Murphy<br>Deryck J Mills<br>Özkan Yildiz<br>Werner Kühlbrandt |
| Deutsche Forschungsge-meinschaft | | Niklas Klusch<br>Werner Kühlbrandt |
| European Molecular Biology Organization | ALTF 702–2016 | Bonnie J Murphy |

The funders had no role in study design, data collection and interpretation, or the decision to submit the work for publication.

### Author contributions

Niklas Klusch, Data curation, Formal analysis, Validation, Investigation, Visualization, Methodology, Writing—original draft, Grew Polytomella cultures and isolated ATP synthase dimers, Prepared cryo-EM specimens and recorded image data, Built the atomic model, Analysed the structure, Drew the figures; Bonnie J Murphy, Data curation, Formal analysis, Validation, Investigation, Visualization, Methodology, Writing—original draft, Prepared cryo-EM specimens and recorded image data, Processed image data and generated the 3D map, Built the atomic model, Analysed the structure; Deryck J Mills, Methodology, Prepared cryo-EM specimens and maintained electron microscopes and cameras; Özkan Yildiz, Resources, Formal analysis, Validation, Visualization, Writing—review and editing, Built the atomic model, Analysed the structure; Drew the figures; Werner Kühlbrandt, Conceptualization, Resources, Formal analysis, Supervision, Funding acquisition, Investigation, Writing—original draft, Project administration, Writing—review and editing, Initiated and directed the study, Analysed the structure

Author ORCIDs

Bonnie J Murphy http://orcid.org/0000-0001-6341-9368

Özkan Yildiz http://orcid.org/0000-0003-3659-2805

Werner Kühlbrandt http://orcid.org/0000-0002-2013-4810

Decision letter and Author response

Decision letter https://doi.org/10.7554/eLife.33274.026

Author response https://doi.org/10.7554/eLife.33274.027

## Additional files

### Major datasets

The following datasets were generated:

| Author(s) | Year | Dataset title | Dataset URL | Database, license, and accessibility information |
|---|---|---|---|---|
| Klusch N, Murphy BJ, Mills DJ, Yildiz Ö, Kühlbrandt W | 2017 | Cryo-EM structure of Polytomella ATP synthase Fo complex | https://www.rcsb.org/pdb/search/structid-Search.do?structureId=6F36 | Publicly available at the RCSB Protein Data Bank (accession no. 6F36) |
| Klusch N, Murphy BJ, Mills DJ, Yildiz Ö, Kühlbrandt W | 2017 | Cryo-EM structure of Polytomella ATP synthase Fo complex | https://www.ebi.ac.uk/pdbe/entry/emdb/EMD-4176 | Publicly available at the EM DataBank (accession no. EMDB-4176) |

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
