## [Decision Letter]

Thank you for submitting your article "Structural basis of proton translocation and force generation in mitochondrial ATP synthase" for consideration by *eLife*. Your article has been favorably evaluated by John Kuriyan (Senior Editor) and three reviewers, one of whom, Sjors HW Scheres (Reviewer #1), is a member of our Board of Reviewing Editors. The following individual involved in review of your submission has agreed to reveal their identity: Henning Stahlberg (Reviewer #3).

The reviewers have discussed the reviews with one another and the Reviewing Editor has drafted this decision to help you prepare a revised submission.

This paper describes a cryo-EM structure to an overall resolution of 4.1A of the intact FoF1 ATP synthase dimer from the alga *Polytomella sp*. Focussed refinement on the dimeric Fo region yielded a map to 3.7A resolution, in which an atomic model could be built. The manuscript focuses on the description of this part of the complex. After a description of the yeast Fo dimer by another group, this manuscript claims to be the first describing the Fo dimer in a functionally competent dimer. The manuscript describes the proton pathway, with detailed descriptions of the half-channels on either side of the c-ring, and provides a mechanism of ring rotation. All three reviewers agreed that this paper represents an exciting step forwards in understanding these molecular machines (with one reviewer envisioning seeing it in future textbooks) and all three reviewers recommended publication.

The following points should be addressed in a revised version:

1) Atomic model validation metrics: this paper needs map-to-model FSC analyses, a molprobity report that includes a clashscore, Ramachandran statistics, and an EMRINGER quantification of rotamer placement. The paper doesn't mention any model refinement approach, but at this resolution, that should improve the statistics, so a model refinement should probably be included in the paper.

2) When describing the mechanism of ring rotation, the authors claim that the field across the sub-nanometer distance between the half-channels "would exert a significant force […], resulting in a net directional ring rotation", but provide no data on how "significant" this force would be. How large will this force be? How significant is it compared to the alternative model of stochastic bidirectional movement? What is missing here is a specifically stated conclusion about which of the two models (stochastic bidirectional movement or potential gradient force) is most likely to be correct.

3) Many of the figures that used a mesh for the cryoEM density, especially the stereo-images, were too busy for easy interpretation. A transparent isosurface, or removal of the density altogether (perhaps while showing zoomed in regions separately in supplementary figures?) may be easier for readers to digest, especially with all of the residue labels.

4) The readability for the wider readership of *eLife* may be improved by adding a figure with an overview sketch, showing the enzyme subunits, their names, and where they are placed in the entire dimeric structure. Something along the lines of Figure 6 in this manuscript http://www.plantphysiol.org/content/144/2/1190.long would do it.

Please find the individual reviewers' comments below. It would probably be beneficial to the paper if most of the other comments are also incorporated, but provided the ones above are addressed, the manuscript will most likely be ready for acceptance.

*Reviewer #1:*

This paper describes a cryo-EM structure to an overall resolution of 4.1A of the intact FoF1 ATP synthase dimer from the alga *Polytomella sp*. Focussed refinement on the dimeric Fo region yielded a map to 3.7A resolution, in which an atomic model could be built. The manuscript focuses on the description of this part of the complex. After a description of the yeast Fo dimer by another group, this manuscript claims to be the first describing the Fo dimer in a functionally competent dimer. The manuscript describes the proton pathway, with detailed descriptions of the half-channels on either side of the c-ring, and provides a mechanism of ring rotation. Overall, the findings are interesting and worthy of publication in *eLife*, provided the following criticism is taken into account:

In the cover letter the authors describe that the F1 heads adopt multiple conformations and that more data is needed. This statement seems to conflict with the one that most side chains in the dimer are visible. It would be better to include a statement in the main text on the heterogeneity in the F1 heads to explain why the manuscript focuses on the Fo monomer structure.

Wouldn't it be interesting to also comment on the dimerisation interface of the Fo subunits? Both in the membrane and in the stalk region? The yeast manuscript describes these interfaces in detail. Does this interface change in this functionally competent dimer? In general, it would be interesting to know what are the differences and similarities with the yeast structure.

When describing the mechanism of ring rotation, the authors claim that the field across the sub-nanometer distance between the half-channels "would exert a significant force […], resulting in a net directional ring rotation", but provide no data on how "significant" this force would be. How large will this force be? How significant is it compared to the alternative model of stochastic bidirectional movement?

The atomic model was built and fitted manually in Coot. However, at this resolution, it should also be refined (e.g. using Phenix or Refmac). This will improve the fit to the density, and the resulting statistics about the fit (please insert FSC model-vs-map curves) and the geometry (please include MolProbity and perhaps EM-ringer stats) will provide the reader with a means to assess its quality.

*Reviewer #2:*

The report by Klusch et al. is an exciting step forward in our understanding of the mechanism of proton motive force-driven rotary ATP catalysis that I would like to see published without undue delay. I can easily imagine seeing their Figure 6 reproduced in textbooks in the near future. The key message of this paper concerns the path of protons through F1F0 complex of a green alga to drive rotation by directional protonation and deprotonation of glutamate residues found in the c-ring. The proton path depends on the fascinating arrangement of a helical hairpin found in the α subunit stator. In addition, these authors identify a critical Arg residue "seal" that prevents leaking between adjacent aqueous paths and provide new insight into how the complex dimerizes to bend the cristae membrane. Of course, similar insights were recently described in Guo et al. (Science Oct 26th) based on a similar structure of the yeast F1F0 complex. Therefore, I hope to see this paper published in 2017 if possible, provided the authors can redress some deficiencies:

1) Atomic model validation metrics.

I would like to see map-to-model FSC analyses, a molprobity report that includes a clashscore, Ramachandran statistics, and an EMRINGER quantification of rotamer placement.

*Reviewer #3:*

The authors describe the 3.7A structure of an entire F-ATPase dimer. Their high resolution allows resolving the aqueous proton access channels through the a-subunit, and quantitatively explain the pathway of protons onto the c-ring, and which forces are involved in driving the enzyme.

The manuscript is nicely written, has very impressive data and findings, and is an important progress in the field of F-ATPase. The figures are clear, beautiful, and contribute significantly to the understandability of the manuscript.

However, manuscript may be a bit specialized for the general life sciences readership of *eLife*. To make the Results section and the Discussion more understandable to the general audience, the authors should include an overview sketch, showing the enzyme subunits, their names, and where they are placed in the entire dimeric structure. Something along the lines of Figure 6 in this manuscript http://www.plantphysiol.org/content/144/2/1190.long would do it.

In the last paragraph of the subsection “Structure determination and atomic model”, the authors first mention "the 11-residuel loop connecting H4 and H5", and only later introduce the a-subunit as being composed of six alpha-helices termed H1 to H6. The order here should be inverted.

In the subsection “Mechanism of ring rotation”, the authors discuss two different rotation models. In model 1 by Junge et al. 1997 of a stochastic bidirectional movement that is biased towards one direction by a proton-motive force (i.e., chemical plus electric gradient force), which is also called a ratchet motor, I believe. In model 2 by Miller et al., 2013, a net electrostatic field pulls a negatively charged glutamate forward, cause direct rotation, which may be called a power stroke.

The authors then calculate that the electric potential of 200mV over the short distance between the two aqueous channels onto the C-ring would correspond to 100'000'000 Volt/m perpendicularly to the membrane normal, and that this would exert a significant force, resulting in net directional rotation. What is missing here is a specifically stated conclusion. I interpret the author's statement as that they argue to have proven Model 2 above. However, this statement is missing.

Also, the force by 100e6 V/m on one charge with the given C-ring radius would have to be compared to the force of Brownian motion on the entire C-ring, in order to be able to state that a purely electrostatic power stroke mechanism drives the ring, and not biased Brownian motion. A quantitative comparison, and a clear final statement are missing.

In the Materials and methods section, the authors describe that they first used Unblur, and then followed by MotionCor2 with magnification distortion correction, which changed the pixel size from 1.12 to 1.105 A. This is a very unusual motion correction, which should accumulate interpolation errors from the two drift correction runs. The authors could add one sentence, explaining why they used two consecutive motion correction procedures, and why they believe the iterative drift correction was required.

---

## [Author Response]

The following points should be addressed in a revised version:1) Atomic model validation metrics: this paper needs map-to-model FSC analyses, a molprobity report that includes a clashscore, Ramachandran statistics, and an EMRINGER quantification of rotamer placement. The paper doesn't mention any model refinement approach, but at this resolution, that should improve the statistics, so a model refinement should probably be included in the paper.

The model-to-map FSC is now included in Figure 1—figure supplement 2. We uploaded the PDB validation report and EM map deposition as supplementary information in the *eLife* submission system. The clashscore, Ramachandran statistics and EMRinger score are now shown in the new Table 1 that summarizes the cryoEM data acquisition, data processing and model statistics. The final model has been refined using PHENIX (phenix.real_space_refine). The procedure is described in the Materials and methods section of the revised manuscript.

2) When describing the mechanism of ring rotation, the authors claim that the field across the sub-nanometer distance between the half-channels "would exert a significant force […], resulting in a net directional ring rotation", but provide no data on how "significant" this force would be. How large will this force be? How significant is it compared to the alternative model of stochastic bidirectional movement? What is missing here is a specifically stated conclusion about which of the two models (stochastic bidirectional movement or potential gradient force) is most likely to be correct.

We are happy to include a quantitative estimate of the force exerted on the negatively charged *c*-ring glutamate, as requested by the reviewers. The “significant force” in vacuum would be 53 pN. At the 2.1 nm radius of the glutamate sidechain, the resulting torque is 112 pN nm. Since the protein dielectric would reduce the local field strength to less than 50% of that in vacuum, the net torque is somewhere around 50 pN nm, in remarkably good agreement with the torque generated by the *E. coli* F_1_ ATPase, determined experimentally as 40 to 60 pN nm by three independent groups. This is now explained in the revised Discussion subsection “Mechanism and energetics of ring rotation” (second paragraph), where the three papers reporting the experimentally determined F_1_ torque are cited. Since the basic mechanisms of torque generation and ATP synthesis are highly conserved and essentially identical in rotary ATPases of mitochondria, chloroplasts and bacteria, torque measurements made with the *E. coli* ATP synthase are relevant for the mitochondrial ATP synthase as well.

The stochastic bidirectional model does not predict an explicit force or torque acting on the *c*-ring glutamates. As the two models are not mutually exclusive, it would be wrong to say that one of them is incorrect. Even if the ring moves back and forth stochastically, the local electrostatic field across the channels we describe would provide a strong bias towards ring rotation in the experimentally observed counter-clockwise direction (as seen from the matrix) in ATP synthesis mode.

3) Many of the figures that used a mesh for the cryoEM density, especially the stereo-images, were too busy for easy interpretation. A transparent isosurface, or removal of the density altogether (perhaps while showing zoomed in regions separately in supplementary figures?) may be easier for readers to digest, especially with all of the residue labels.

The 3D structure of the F_o_ ATP synthase is fairly complex and not easy to display in two-dimensional figures. We tried to do our best but realized the result was not perfect. Presumably, anyone interested in the details of the structure and mechanism will look at the three-dimensional map and model with a suitable molecular graphics program, once the pdb coordinates are released, which they will be on acceptance of the manuscript. Nevertheless we have simplified Figure 4 as requested, which is now non-stereo and the map has been omitted (except for the channel surfaces). The original figure with the map is now Figure 4—figure supplement 1. A supplementary movie for Figure 4—figure supplement 3 was added (Video 3 and Video 4).

4) The readability for the wider readership of eLife may be improved by adding a figure with an overview sketch, showing the enzyme subunits, their names, and where they are placed in the entire dimeric structure. Something along the lines of Figure 6 in this manuscript http://www.plantphysiol.org/content/144/2/1190.long would do it.

Thank you for this suggestion. The new overview sketch is Figure 1—figure supplement 1.

Please find the individual reviewers' comments below. It would probably be beneficial to the paper if most of the other comments are also incorporated, but provided the ones above are addressed, the manuscript will most likely be ready for acceptance.Reviewer #1:[…] Overall, the findings are interesting and worthy of publication in eLife, provided the following criticism is taken into account:In the cover letter the authors describe that the F1 heads adopt multiple conformations and that more data is needed. This statement seems to conflict with the one that most side chains in the dimer are visible. It would be better to include a statement in the main text on the heterogeneity in the F1 heads to explain why the manuscript focuses on the Fo monomer structure.Good point. We have expanded first sentence of the Results: “… in which most sidechains of the ~14,000 residue dimer are visible (Figure 1; Video 1), except in the F_1_ heads and central stalk, where blending of multiple rotational states reduces map definition.”Wouldn't it be interesting to also comment on the dimerisation interface of the Fo subunits? Both in the membrane and in the stalk region? The yeast manuscript describes these interfaces in detail. Does this interface change in this functionally competent dimer? In general, it would be interesting to know what are the differences and similarities with the yeast structure.

It would indeed be interesting, but since the dimerization interfaces of the yeast and *Polytomella* dimers are completely different, a detailed comparison of the dimer interface would go well beyond the scope of the present manuscript and delay its publication unreasonably. With the exception of subunit *a* and the *c*-ring, none of the yeast F_o_ and peripheral stalk subunits are even remotely similar. This is remarkable in itself, since the key mechanism of torque generation appears to be completely conserved. These and other important differences, such as the rotary position of the *c*-ring relative to subunit *a*, will be the subject of another paper to be submitted in due course.

When describing the mechanism of ring rotation, the authors claim that the field across the sub-nanometer distance between the half-channels "would exert a significant force […], resulting in a net directional ring rotation", but provide no data on how "significant" this force would be. How large will this force be? How significant is it compared to the alternative model of stochastic bidirectional movement?

See detailed response to comment 2 above.

The atomic model was built and fitted manually in Coot. However, at this resolution, it should also be refined (e.g. using Phenix or Refmac). This will improve the fit to the density, and the resulting statistics about the fit (please insert FSC model-vs-map curves) and the geometry (please include MolProbity and perhaps EM-ringer stats) will provide the reader with a means to assess its quality.

See detailed response to comment 1 above. The model has been refined and a table of refinement statistics added (Table 1). The model-vs-map FSC has been added to Figure 1—figure supplement 2.

Reviewer #2:[…] I hope to see this paper published in 2017 if possible, provided the authors can redress some deficiencies:1) Atomic model validation metrics.I would like to see map-to-model FSC analyses, a molprobity report that includes a clashscore, Ramachandran statistics, and an EMRINGER quantification of rotamer placement.

See detailed response to comment 1 and corresponding comment of Reviewer 1 above.

Reviewer #3:[…] However, manuscript may be a bit specialized for the general life sciences readership of eLife. To make the Results section and the Discussion more understandable to the general audience, the authors should include an overview sketch, showing the enzyme subunits, their names, and where they are placed in the entire dimeric structure. Something along the lines of Figure 6 in this manuscript http://www.plantphysiol.org/content/144/2/1190.long would do it.

Thank you for this suggestion. We have added a schematic overview of the *Polytomella* ATP synthase dimer as Figure 1—figure supplement 1.

In the last paragraph of the subsection “Structure determination and atomic model”, the authors first mention "the 11-residuel loop connecting H4 and H5", and only later introduce the a-subunit as being composed of six alpha-helices termed H1 to H6. The order here should be inverted.

Done, thank you.

In the subsection “Mechanism of ring rotation”, the authors discuss two different rotation models. In model 1 by Junge et al. 1997 of a stochastic bidirectional movement that is biased towards one direction by a proton-motive force (i.e., chemical plus electric gradient force), which is also called a ratchet motor, I believe. In model 2 by Miller et al., 2013, a net electrostatic field pulls a negatively charged glutamate forward, cause direct rotation, which may be called a power stroke.The authors then calculate that the electric potential of 200mV over the short distance between the two aqueous channels onto the C-ring would correspond to 100'000'000 Volt/m perpendicularly to the membrane normal, and that this would exert a significant force, resulting in net directional rotation. What is missing here is a specifically stated conclusion. I interpret the author's statement as that they argue to have proven Model 2 above. However, this statement is missing.Also, the force by 100e6 V/m on one charge with the given C-ring radius would have to be compared to the force of Brownian motion on the entire C-ring, in order to be able to state that a purely electrostatic power stroke mechanism drives the ring, and not biased Brownian motion. A quantitative comparison, and a clear final statement are missing.See detailed response to comment 2 above.In the Materials and methods section, the authors describe that they first used Unblur, and then followed by MotionCor2 with magnification distortion correction, which changed the pixel size from 1.12 to 1.105 A. This is a very unusual motion correction, which should accumulate interpolation errors from the two drift correction runs. The authors could add one sentence, explaining why they used two consecutive motion correction procedures, and why they believe the iterative drift correction was required.A sentence to this effect has been added to the Materials and methods of the revised manuscript: “Combining these two programs gave better results than either program alone, as judged by quality of a 3D reconstruction for a subset of the micrographs.”